# Optimized Heterologous Expression and Efficient Purification of a New TRAIL-Based Antitumor Fusion Protein SRH–DR5-B with Dual VEGFR2 and DR5 Receptor Specificity

**DOI:** 10.3390/ijms23115860

**Published:** 2022-05-24

**Authors:** Anne V. Yagolovich, Artem A. Artykov, Alina A. Isakova, Yekaterina V. Vorontsova, Dmitry A. Dolgikh, Mikhail P. Kirpichnikov, Marine E. Gasparian

**Affiliations:** 1Shemyakin-Ovchinnikov Institute of Bioorganic Chemistry of the Russian Academy of Sciences, 117997 Moscow, Russia; art.al.artykov@gmail.com (A.A.A.); alina.labbio@gmail.com (A.A.I.); kate.voronts09@gmail.com (Y.V.V.); dolgikh@nmr.ru (D.A.D.); kirpichnikov@inbox.ru (M.P.K.); 2Faculty of Biology, Lomonosov Moscow State University, 119234 Moscow, Russia

**Keywords:** bifunctional fusion proteins, fusion expression, *E. coli*, TRAIL DR5-selective variant DR5-B, DR5 receptor, VEGFR2 receptor

## Abstract

In the last two decades, bifunctional proteins have been created by genetic and protein engineering methods to increase therapeutic effects in various diseases, including cancer. Unlike conventional small molecule or monotargeted drugs, bifunctional proteins have increased biological activity while maintaining low systemic toxicity. The recombinant anti-cancer cytokine TRAIL has shown a limited therapeutic effect in clinical trials. To enhance the efficacy of TRAIL, we designed the HRH–DR5-B fusion protein based on the DR5-selective mutant variant of TRAIL fused to the anti-angiogenic synthetic peptide HRHTKQRHTALH. Initially low expression of HRH–DR5-B was enhanced by the substitution of *E. coli*-optimized codons with AT-rich codons in the DNA sequence encoding the first 7 amino acid residues of the HRH peptide. However, the HRH–DR5-B degraded during purification to form two adjacent protein bands on the SDS-PAGE gel. The replacement of His by Ser at position P2 immediately after the initiator Met dramatically minimized degradation, allowing more than 20 mg of protein to be obtained from 200 mL of cell culture. The resulting SRH–DR5-B fusion bound the VEGFR2 and DR5 receptors with high affinity and showed increased cytotoxic activity in 3D multicellular tumor spheroids. SRH–DR5-B can be considered as a promising candidate for therapeutic applications.

## 1. Introduction

Most pharmaceuticals based on recombinant proteins are produced in microbial host cells such as bacteria or yeast [1]. This indicates that the microbial systems are convenient and powerful tools for the production of recombinant proteins, despite the various bottlenecks and obstacles that they pose for the efficient production of functional mammalian proteins, namely the absence or unconventional post-translational modifications, proteolytic instability, poor solubility, etc. [2]. In addition, animal cell culture media are about 100–1000 times more expensive than bacterial or fungal culture media [3]. The cytokine TRAIL (TNF-related apoptosis-inducing ligand) from the TNF family is a promising anticancer drug that induces the apoptosis of cancer cells while leaving normal cells intact [4]. All abbreviations used in this study are listed in back matter. In humans, TRAIL triggers an apoptotic signal through two receptors, TRAIL-R1 and TRAIL-R2, also known as the death receptors DR4 and DR5, respectively [5]. The soluble recombinant human TRAIL expressed and purified from bacterial strains has induced marked tumor regression without systemic toxicity in animal models, and this has led to the clinical development of novel drugs targeting death receptors DR4 and DR5 [6,7]. Despite high antitumor activity in animal models, TRAIL showed very limited antitumor properties in clinical trials [8]. Over the past decade, many TRAIL derivatives with improved agonistic potential have been developed to increase its antitumor efficacy [9]. In particular, a number of bifunctional TRAIL-based fusion proteins have been designed to simultaneously target death receptors and the receptors associated with proliferative signaling or angiogenesis [10]. Usually TRAIL and its derivatives were expressed by standard IPTG induction in *E. coli* strains BL21(DE3) [11,12], SHuffle^®^ T7 Express [13], M15 [14] or TG1 [15]. Furthermore, expression by autoinduction in *E. coli* strain BL21(DE3) has been described for the recombinant fusions IL2-TRAIL [16] and ADI (arginine deiminase)-TRAIL [17]; however, no information on protein yield has been provided. Eukaryotic expression has been commonly used to express TRAIL fused with antibodies that are poorly produced in the prokaryotic expression systems. CHO cells have been used for the production of several TRAIL fusion proteins with antibodies (scFv:sTRAIL), including proteins with specificity for EpCAM and EGFR solid tumor antigens and various leukemia target antigens (CD7, CD19, CD33) [18]. Mammalian cells such as CHO or HEK293 have been used to generate a single polypeptide chain variant of TRAIL (scTRAIL) consisting of three monomers fused by short peptide linkers, which was further fused with antibodies (scFv-scTRAIL) [19,20,21]. Among more than five dozen TRAIL fusion proteins, only ABBV-621, a single chain TRAIL trimer dimerized via a modified Fc part, is currently in clinical trials for the treatment of solid and hematologic malignancies [22].

The activation of angiogenesis is one of the first requirements in the tumor ecosystem for the growth and dissemination of tumors and stromal cells [23]. VEGF is a potent angiogenic agent in tumor tissues, and the role of VEGF receptors (VEGFRs) has been extensively studied in the field of neoplastic vascularization [24]. However, agents targeting the VEGF/VEGFR pathway, such as bevacizumab, sorafenib and sunitinib, have side effects including hypertension, arterial thromboembolic events, renal dysfunction, wound complications, bleeding, gastrointestinal perforation and reversible posterior leukoencephalopathy syndrome [25]. Therefore, the combination of selective inhibition of angiogenesis with cancer-cell-specific antitumor agents is a promising therapeutic strategy. TRAIL gains attention because its target receptors DR4 and DR5 are expressed much higher in cancer cells compared to non-transformed cells. Recently, a bifunctional recombinant TRAIL fused with an anti-angiogenic effector peptide repetitive sequence derived from exon 6 of the VEGF ligand demonstrated strong antitumor activity in xenograft models [26].

In current work, we report the efficient heterologous high-yield production in *E. coli* of a novel bifunctional protein HRH–DR5-B based on DR5-selective TRAIL mutant variant DR5-B genetically fused with an earlier reported synthetic anti-angiogenic peptide HRH (HRHTKQRHTALH) [27]. However, the addition of an extra HRH peptide to the N-end of DR5-B hindered protein stability, as the HRH–DR5-B fusion protein was degraded during purification to form two adjacent protein bands on the SDS-PAGE gel. The substitution of the amino acid residue His to Ser at the P2 position drastically reduced protein degradation, allowing the obtention of more than 20 mg of the highly purified fusion protein from 200 mL of the cell culture. The resulting SRH–DR5-B fusion retained high affinity to both the VEGFR2 and DR5 receptors and demonstrated increased cytotoxic activity in 3D multicellular tumor spheroids, so it can be considered a prospective candidate for concurrent antitumor and antiangiogenic therapy.

## 2. Results

### 2.1. Heterologous Expression of HRH–DR5-B Fusion in Escherichia coli Strains

Previously, we have generated a DR5-selective mutant variant of TRAIL that bound to the DR5 receptor with an affinity similar to the wild-type cytokine. DR5-B did not bind to the DR4 and DcR1 receptors, and its affinity for the DcR2 and OPG receptors was 400-fold lower compared to that of TRAIL [28]. The expression of DR5-B in the *E. coli* SHuffle B strain transformed with pET32a/sdr5-b plasmid was described earlier [29]. To express the HRH–DR5-B fusion, the DNA encoding the anti-angiogenic HRH (HRHTKQRHTALH) peptide and the GGAAGGAGSGG linker were introduced into the plasmid vector pET32a/sdr5-b immediately before the DR5-B gene sequence (Figure 1A). The DNA sequence encoding the HRH peptide was designed using *E. coli* codon bias with high-frequency codon selection (Figure 1B). However, the level of protein expression was very low in all three tested *E. coli* strains (SHuffle B, BL23(DE3), or BL23(DE3)pLysS, both upon IPTG induction or autoinduction (Figure 1C)). Previously, it was shown that GFP protein expression was inversely correlated with the content of GC nucleotide residues, particularly in 2–7 codons, and the probability of base pairing was predicted [30]. We decided to apply this strategy by replacing the nucleotide residues GC with AT in the codons of the first eight amino acid residues (Figure 2B). These substitutions drastically increased HRH–DR5-B expression in all *E. coli* strains despite having the rare “aga” codon for arginine at positions 3 and 8 (Figure 1C). 

A 14-fold improvement in protein expression was previously reported in *E. coli* with the N-terminal rare codons instead of common ones [31]. The highest protein expression was detected in the SHuffle B strain upon IPTG induction. The expression level of HRH–DR5-B was comparable to that of DR5-B. However, during autoinduction, the protein was better expressed in the BL21(DE3) strains. Therefore, SHuffle B cells with IPTG induction were used to express the target proteins for further purification.

### 2.2. His/Ser Substitution at P2 Position after N-terminal Met Prevents the Degradation of HRH–DR5-B

The HRH–DR5-B fusion was purified using a similar approach as previously described for DR5-B [29]. However, during purification, the protein was steadily degraded, forming two closely spaced bands on the polyacrylamide gel (Figure 2A). We hypothesized that degradation may occur as a result of the retention of the N-terminal Met (methionine), since it is shown not to be processed when His (histidine) is in the P2 position [32]. 

It has previously been shown that the retention of N-Met can lead to the rapid degradation of several proteins that normally undergo N-terminal methionine excision (NME) [33]. NME in bacteria is effective when the amino acids Gly, Ala, Pro, Ser, Thr, Val and Cys are in position P2 [34]. Among them, the polar uncharged Ser (serine) was chosen to replace His at the P2 position by site-directed mutagenesis, and the expression and purification of SRH–DR5-B was performed similarly to HRH–DR5-B. The His/Ser substitution virtually prevented protein degradation (Figure 2A). The yield of purified proteins DR5-B, HRH–DR5-B and SRH–DR5-B averaged 22 mg per 200 mL of cell culture. On the SEC column, recovered SRH–DR5-B, as well as DR5-B, were separated as a single-protein peak of approximately 66 and 60 kDa, respectively (Figure 2B). The calculated mass of monomeric DR5-B and SRH–DR5-B proteins was 19,624.04 Da and 21,769.42 Da, respectively. Accordingly, the molecular weight of trimeric DR5-B and SRH–DR5-B molecules was 58.872 kDa and 65.308 kDa, respectively. These data indicated that both proteins form homogenous trimers.

### 2.3. SRH–DR5-B Showed High Affinity for DR5 and VEGFR2 Receptors

To analyze the effect of the His/Ser substitution on receptor affinity, the dissociation constants (*KD*) of the HRH–DR5-B and SRH–DR5-B protein preparations for the VEGFR1 and VEGFR2 receptors were calculated. The binding affinity and static binding capacity of proteins to receptors was quantified as an equilibrium dissociation constant (*KD*) by an enzyme-linked immunosorbent assay (ELISA) (Figure 3A).

Expectedly, DR5-B itself had no affinity for VEGFR2 (data not shown). The affinity of SRH–DR5-B to the VEGFR2 receptor (*KD* = 1.2 ± 0.09 nM) was higher in comparison to HRH–DR5-B (*KD* = 3.73 ± 0.17 nM), which can be explained by the partial degradation of HRH–DR5-B fusion. Thus, the fusion protein SRH–DR5-B binds to the VEGFR2 receptor in the nanomolar concentration range, which is comparable to the affinity of the natural VEGFA ligand with *KD* = 0.15–170 nM depending on the measurement method [35]. The synthetic anti-angiogenic peptide HRHTKQRHTALH was previously isolated from a phage display library using VEGFR-Fc as a bait [27]. The peptide did not lose its affinity for the VEGFR2 receptor after fusion with DR5-B at the N-terminus. This is probably due to the linker (GGAAGGAGSGG) between HRH and DR5-B, which provides free access of the peptide to the receptor. The HRH–DR5-B and SRH–DR5-B fusion proteins showed almost no affinity to the VEGFR1 receptor (data not shown), which is a negative regulator of VEGFA signaling to control blood vessel growth and morphogenesis [36].

To ensure that the addition of the *N*-terminal SRH peptide does not impair DR5 receptor binding, a comparative analysis of DR5-B and SRH–DR5-B proteins for DR5 binding was carried out by ELISA (Figure 3B). The affinity of SRH–DR5-B to the DR5 receptor was comparable to that of DR5-B (*KD* = 1.19 ± 0.07 nM and 1.74 ± 0.03 nM for DR5-B and SRH–DR5-B, respectively). It should be noted that the *KD* values of the DR5-B protein for the DR5 receptor, obtained by ELISA, almost coincided with our previous results obtained by the surface plasmon resonance (0.71 ± 0.013 nM) [28].

### 2.4. The Cytotoxic Activity of SRH–DR5-B in 2D and 3D Tumor Cell Models

The ability of SRH–DR5-B to reduce tumor cell viability was evaluated in tumor cells of different origin, namely glioblastoma (T98G and U-87) and colorectal carcinoma (HT-29 and HCT116) cell lines. Flow cytometry analysis showed that all lines express the DR5 receptor on the cell surface (Figure 4A). SRH–DR5-B significantly suppressed the viability of the HCT116, HT-29 and T89G cells after 24 h and 48 h, similar to the activity of DR5-B, as determined by WST-1 assay (Figure 4B). In U-87 cells, SRH–DR5-B was less effective than DR5-B after 24 h (IC50 values 1.62 ± 0.28 nM and 8.62 ± 1.25 nM for DR5-B and SRH–DR5-B, respectively). Interestingly, with a longer incubation (48 h), the fusion protein worked more efficiently than DR5-B (IC50 values 0.68 ± 0.11 nM and 0.056 ± 0.012 nM for DR5-B and SRH–DR5-B, respectively). In U-87 cells, the prolonged (more than 24 h) incubation with a potent and selective inhibitor of VEGFR2 kinase activity SU1498 has been previously shown to cause a significant increase in the percentage of cells with late apoptosis [37]. Probably, SRH–DR5-B inhibits the activity of VEGFR2 in a time-dependent manner, causing more efficient cell death during prolonged incubation.

Tests on two-dimensional (2-D) monolayer cell cultures cannot always be satisfactorily confirmed in animal studies or in clinical trials, because of the inability to mimic the extracellular microenvironment where cells reside in tumor tissues. Therefore, we compared the cytotoxic activity of SRH–DR5-B and DR5-B by a WST-1 test in three-dimensional structures (3D) of HT-29, T98G and U-87 cells (300–400 µM in size), using a classical technique with agarose hydrogel as a polymeric scaffold, as agarose is a natural polysaccharide polymer with excellent biocompatibility [38]. SRH–DR5-B was more effective in HT-29 and U-87 cells, and the effect increased with prolonged incubation for 48 h (Figure 5A). The IC50 of SRH–DR5-B was 14 and 3.5 times lower for HT-29 and U-87 cells, respectively, compared to DR5-B (Figure 5B). Additionally, SRH–DR5-B induced higher maximal cell death in U-87 cells. In the T98G cell spheroids, the differences in IC50 values of the two ligands also decreased after 48 h of incubation (Figure 5B). Images of the cell spheroids obtained by the light microscopy clearly reflect the results of the WST-1 cell viability test (Figure 5C). The enhanced efficiency of SRH–DR5-B is obviously related to the level of the surface expression of VEGFR2 and DR5 receptors, which may differently affect the dynamics of the receptor activation.

## 3. Discussion

Modern bioengineering approaches make it possible to create bifunctional proteins (BFPs) of various types for targeted cancer therapy, which simultaneously activate or block two signaling pathways [39]. TRAIL represents one of the most promising cancer treatments due to its specificity, safety and encouraging results in vitro and in vivo. However, its complex interactions with receptors in a cell-type-dependent manner, along with the resistance of many types of tumors, pose significant barriers to the clinical development of TRAIL-based therapeutics. Despite the lack of efficacy in previous clinical trials, new, more effective TRAIL-based drugs are being developed, in particular bifunctional fusion molecules to improve drug stability and pharmacokinetics and/or suppress tumor proliferative activity [40]. 

As the vascular endothelial growth factor (VEGF) is a potent mitogenic factor in vasculogenesis and angiogenesis, the blockade of VEGF-mediated signals may also prevent tumor growth by enhancing tumor cell apoptosis. Consequently, therapeutic intervention aimed at inhibiting the VEGFR2 pathway has become the mainstay of cancer treatment [41]. Here, we have described the heterologous production of a bifunctional recombinant fusion protein in *E. coli* based on a DR5-selective mutant variant of TRAIL DR5-B equipped with the N-terminal synthetic anti-angiogenic peptide HRHTKQRHTALH (HRH), which was previously isolated from a phage display library using a VEGFR-Fc as bait [27]. It was assumed that generation of a bifunctional protein simultaneous targeting the DR5 and VEGFR2 receptors would enhance the antitumor activity of DR5-B.

*E. coli* is the most widely used host for the biopharmaceutical heterologous production of recombinant proteins due to its simplicity, speed and low cost. However, the codon bias discrepancy can seriously interfere with protein expression in *E. coli* [42]. The codon optimization strategy commonly used for high-level heterologous protein expression in microbial expression systems [43] resulted in extremely low expression of the HRH–DR5-B fusion in *E. coli* strains. So we replaced the “cgt” codon for arginine at positions +3 and +8 with the rare “aga” codon, as using N-terminal rare codons instead of common ones has been shown to increase expression in *E. coli* by about 14-fold [31]. This strategy, together with the enrichment of the DNA sequence with adenine (A) and thymine (T) nucleotides in the first seven codons, led to a significant increase in the expression level of HRH–DR5-B fusion comparable to that of DR5-B. As a result, an average of 22 mg of DR5-B and HRH–DR5-B was purified per 200 mL of cell culture with a purity of 98%. Besides the expression, another challenge in the heterologous production of recombinant fusion proteins is the correct folding and stability of the purified protein. We encountered this during the purification of HRH–DR5-B, which was steadily degraded to form two closely spaced bands on a polyacrylamide gel. Possibly, the addition of an artificial polypeptide to the *N*-terminus of DR5-B led to the destabilization and misfolding of the fusion protein. We also suggested that degradation may occur as a result of the retention of *N*-terminal methionine in the presence of a.a. His at position P2 after the Met initiator [32], since the retention of Met can lead to rapid degradation of proteins [33]. N-terminal methionine excision (NME) in bacteria is effective when the amino acids Gly, Ala, Pro, Ser, Thr, Val and Cys are in position P2 [34]. It was also shown that a.a. Ala, Pro, Ser, Thr, Val, Cys and Lys at position P2 resulted in the significantly higher expression of recombinant Igα than other amino acids [44]. Among them, the polar uncharged Ser (serine) was chosen to replace His at the P2 position. As a result, the His/Ser substitution did not affect the expression level and virtually prevented protein degradation. 

The size-exclusion chromatography analysis clearly showed that purified SRH–DR5-B, same as DR5-B, consists of homogeneous trimeric molecules. The affinity of SRH–DR5-B to the VEGFR2 receptor was three times higher compared to HRH–DR5-B, which may be due to the partial degradation of the latter fusion. SRH–DR5-B binds to the VEGFR2 receptor at nanomolar concentrations that are comparable to the affinity of native VEGFA ligand [35]. In addition, SRH–DR5-B bound to the DR5 receptor as efficiently as DR5-B, which made it possible to compare their cytotoxic activity in 2D and 3D tumor cell models. 

In monolayer cell cultures, the cytotoxic activity of SRH–DR5-B was comparable to that of DR5-B. In 3D tumor spheroids of the HT-29 and U-87 cell lines, the efficacy of SRH–DR5-B was significantly higher than that of DR5-B, and this effect was more pronounced after a longer incubation period. The difference in sensitivity between the 2D and 3D models may be due to modifications of the expression of growth factors and cytokines, also caused by the ligands themselves. For example, in a 3D culture of Ea.hy926 cells, the production of both pro-inflammatory and anti-inflammatory factors was increased compared to the 2D culture [45]. In addition, the TNF-induced secretion of IL-10, GM-CSF and IL-6 was higher in 3D cultures compared to 2D. Evidence is accumulating indicating that multiple signals from different components of the tumor microenvironment simultaneously affect TRAIL receptor signaling [46]. Despite a number of studies on the concurrent targeting of TRAIL and VEGF signaling pathways, the effect of angiogenesis, and in particular VEGFR2 signaling, on the anticancer activity of TRAIL has been poorly studied. For example, dual mechanistic TRAIL nanocarrier (PEG-LHT7/TRAIL/Protamine) based on PEGylated heparin taurocholate and protamine, which exerts both pro-apoptotic, and anti-angiogenic effects, demonstrated higher antitumor activity than a combination of PEG-LHT7/protamine and TRAIL [47]. The VAS–TRAIL fusion protein, consisting of the angiogenesis-inhibiting fragment of calreticulin vasostatin and TRAIL, demonstrated higher apoptotic activity in endothial FBHE cells than either TRAIL or vasostatin alone [48]. Therefore, it is believed that the use of multifunctional molecules will be a more effective therapeutic strategy than just a combination of drugs. Of course, any bifunctional protein preparation may have potential limitations or problems in clinical use. BFPs that bind to different receptors on tumor cells or block two signaling pathways at the same time are intended primarily for the treatment of patients with solid tumors, but most of them have not yet been approved for clinical use. Limitations may be related to various factors, such as drug yield and cost, specificity, half-life, ability to penetrate the tumor and adverse effects. The technology we developed provides a high yield of the SRH–DR5-B protein, which demonstrated a high specificity for the DR5 and VEGFR2 receptors. Further preclinical studies will reveal the antitumor efficacy, as well as the potential limitations of the drug for further movement into clinical trials.

## 4. Materials and Methods

### 4.1. Reagents and Cell Lines

Dimethylsulfoxide (DMSO), IPTG (isopropylthio-β-galactoside), ampicillin and WST-1 reagent were obtained from Sigma-Aldrich (St. Louis, MO, USA). All other chemicals were purchased from Applichem (Darmstadt, Germany) unless otherwise specified. All solvents and components of buffer solutions were of analytical grade and used as received. Monoclonal antibodies to TRAIL (MAB375) were from R&D systems (Minneapolis, MN, USA), and anti-DR5 monoclonal antibodies (DR5-01-1) and secondary antibodies Dylight 488 were from GeneTex (Irvine, CA, USA). Bacteria were cultivated using Gibco Bacto yeast extract and Gibco Bacto tryptone (Thermo Fisher Scientific, Waltham, MA, USA). Human colorectal carcinoma HCT116 and HT29 and human glioblastoma U-87 and T98G cell lines were from ATCC (Washington, DC, USA). Cell culture media (DMEM, RPMI1640), 0.05% trypsin-EDTA solution and phosphate-buffered saline tablets were from PanEco (Moscow, Russia). Fetal bovine serum was from HyClone (Cramlington, UK).

### 4.2. Construction of Plasmid Vectors for the Expression of HRH–DR5-B and SRH–DR5-B Fusions

Previously, we constructed the pET32a/sdr5-b vector plasmid for the direct expression of the DR5-B protein [1]. The plasmid vector for the expression of the HRH–DR5-B fusion was designed by inserting the DNA sequence agtagacatacaaaaaaagacaacactgcactccatggtggagcagcaggaggtgctgggagtggtggc, encoding the HRHTKQRHTALH peptide and the GGAAGGAGSGG peptide as a polylinker into the plasmid vector pET32a/sdr5-b right before the DNA encoding the DR5-B protein. The pET32a/hrh–dr5-b plasmid vector for the expression of the HRH–DR5-B fusion protein was ordered from Evrogen (Moscow, Russia). To improve the expression of the HRH–DR5-B fusion protein, GT-rich codons encoding the N-terminal amino acid residues of the HRH peptide were replaced with AT-rich codons using site-directed mutagenesis. The resulting plasmid vector construct was designated pET32a/hrh–dr5-b-at. The HRH–DR5-B fusion construct and the DNA sequences encoding the HRH peptide with optimized and AT-rich codons are shown in Figure 1A,B. To obtain the plasmid vector pET32a/srh-dr5-b-at for the expression of the SRH–DR5-B fusion protein, the codon “cat” for histidine in the plasmid vector pET32a/hrh-sdr5-b-at was changed to “agt” for serine at position P2 immediately after the initiator methionine codon by site-directed mutagenesis.

### 4.3. Expression of Recombinant Proteins in E. coli Strains

For the expression of DR5-B and HRH–DR5-B proteins, the competent cells of *E. coli* strains SHuffle B, BL23(DE3) or BL23(DE3)pLysS were transformed with the plasmid vectors pET32a/dr5-b, pET32a/hrh–dr5-b and pET32a/hrh–dr5-b-at. The colonies were inoculated into the LB (Luria–Bertani) medium in the presence of ampicillin (100 μg/mL), and cell cultures were grown at 37 °C with shaking at 250 rpm for 5 h. Protein expression was induced in two ways. For isopropyl-β-D-1-thiogalactopyranoside (IPTG) induction, the cell culture was diluted (1:100) in TB (terrific broth) medium with ampicillin (100 μg/mL), grown at 37 °C to an optical density of OD600 = 0.6, followed by adding 0.05 mM IPTG. The cell culture was grown at 28 °C for 20 h. Alternatively, the target proteins were expressed by autoinduction. For this, the cell culture was diluted (1:100) in TB supplied by 5052 (0.5% *w/v* glycerol, 0.05% glucose, 0.2% α-lactose), NPS (25 mM (NH_4_)_2_SO_4_, 50 mM KH_2_PO_4_, 50 mM Na_2_HPO_4_), 1 mM MgSO_4_ and ampicillin (100 μg/mL), and the cell culture was grown at 28 °C for 20 h. Cells were sedimented by centrifugation at 5000× *g* (Beckman Coulter, Indianapolis, IN, USA) at 4 °C for 10 min, then washed with a buffer containing 300 mM NaCl and 50 mM Na_2_HPO_4_ (pH 8.0), and the cell pellets were stored at –80 °C for further protein purification.

The expression of the fusion protein SRH–DR5-B was carried out only by the IPTG induction in the *E. coli* SHuffle B strain. The expression level of the target proteins was analyzed by sodium dodecyl sulfate polyacrylamide gel electrophoresis (SDS-PAGE).

### 4.4. Purification of Recombinant Proteins DR5-B, HRH–DR5-B and SRH–DR5-B

The recombinant proteins DR5-B, HRH–DR5-B and SRH–DR5-B were purified as described previously [29]. Briefly, cells were disrupted by French press (Spectronic Instruments Inc., Irvine, CA, USA) under a pressure of 2000 psi, and the proteins were purified from the soluble fraction of the cytoplasmic proteins by immobilized metal-affinity chromatography on Ni-NTA agarose (Qiagen, Germantown, MD, USA), followed by ion exchange chromatography on SP Sepharose (GE Healthcare, Danderyd, Sweden). Purified protein preparations were dialyzed against 50 mM Na_2_HPO_4_ (pH 7.4) and 150 mM NaCl for 24 h at 4 °C, sterilized by filtration, lyophilized and stored at −80 °C.

### 4.5. Size Exclusion Chromatography

The samples containing 1.5 mg of the purified recombinant proteins DR5-B and SRH–DR5-B were applied to gel filtration chromatography in a Superdex 200 10/300 GL column (GE Healthcare, Danderyd, Sweden) equilibrated in PBS at a flow rate of 0.5 mL/min using the AKTA fast protein liquid chromatography (FPLC) system (GE Healthcare, Danderyd, Sweden).

### 4.6. Cell Culture and Multicellular Tumor Spheroids Formation

HT29 cells were cultured in RPMI supplemented with 10% FBS; HCT116, U-87 and T98G cells were cultured in DMEM supplemented with 10% FBS in a 5% CO_2_ humidified atmosphere at 37 °C. The cells were detached after treatment with trypsin–EDTA solution (0.25% *v*/*v*), and the culture medium was replaced every 3–4 days. Multicellular tumor spheroids were produced by seeding the cells (5 × 10^3^ cells/well) in 96-well plates pre-coated with 1.5% agarose. The plates were incubated at 37 °C, and spheroid formation was observed in 48–72 h.

### 4.7. Cytotoxicity Evaluation of Purified Proteins

The cytotoxicity of purified proteins was evaluated by WST-1 assay. Preparations at various dilutions were added to each well, and the cells were transferred to the CO_2_-incubator for 24 or 48 h. Next, 10 µL of WST-1 reagent (Sigma-Aldrich, St. Louis, MO, USA) was added to each well and the plates were incubated for 4 h at 37 °C. The absorbance was measured by an iMark plate spectrophotometer (Bio-Rad, Hercules, CA, USA) at a wavelength of 450 nm with background subtraction at 655 nm. The cell viability was determined in % compared to the control according to the equation: (OD sample—OD background)/(OD control—OD background) × 100%. The half-maximal inhibitory concentration (IC50) was determined to be a drug concentration resulting in the 50% inhibition of cell growth by nonlinear regression in GraphPad Prism 8, (GraphPad Software Inc., San Diego, CA, USA) according to the built-in dose–response inhibition formula.

### 4.8. ELISA

The recombinant extracellular domains of the receptors DR5 (100 ng/well), VEGFR1 or VEGFR2 (50 ng/well) (R&D Systems, Minneapolis, MN, USA) were immobilized on ELISA plates overnight at 4 °C in 0.1 M carbonate–bicarbonate buffer (pH 9.4). The plates were washed three times with PBST (phosphate-buffered saline + 0.05% Tween), and wells were blocked by 2% BSA in PBST for 1 h at 37 °C. After blocking, dilutions of DR5-B, HRH–DR5-B or SRH–DR5-B (in 3 replicates) at concentrations from 0.032 to 2500 nM were added, and the plates were incubated for 1 h at 37 °C. Captured ligands were detected by subsequent incubation with monoclonal antibodies to TRAIL (MAB375, R&D Systems, Minneapolis, MN, USA) and anti-mouse polyclonal goat IgG conjugated with horseradish peroxidase (HAF007, R&D Systems). Unbound antibodies were washed 3 times with a PBST buffer, and color was developed by OPD (o-phenylenediamine dihydrochloride) colorimetric substrate. After 15 min of incubation with the substrate at 37 °C, the reaction was stopped by a 1 N H_2_SO_4_ solution. The optical density was determined at 450 nm by iMark reader (Bio-Rad, Hercules, CA, USA). Dissociation constant (K_D_) values were calculated by GraphPad Prism 8 software (GraphPad Software Inc., San Diego, CA, USA), using the nonlinear regression option in the XY analysis section.

### 4.9. Flow Cytometry

The assays were performed as described earlier, with some modifications [49]. The cells were seeded in a 6-well plate at a density of 2 × 10^5^ cells per well in 2 mL of culture media and incubated for 24 h in a humidified atmosphere of 5% CO_2_ at 37 °C. Cells were detached from the culture flasks with Versene solution, washed with ice-cold PBS and resuspended in FACS buffer (PBS with 1% BSA). Cell suspensions were incubated for 1 h at 4 °C with 5 µg/mL anti-DR5 monoclonal antibodies (clone DR5-01-1, GeneTex, Irvine, CA, USA). Next, the cells were washed twice and incubated with secondary antibodies (20 µg/mL) Dylight 488 (GeneTex Inc.) for 1 h at 4 °C. Cells were washed twice and suspended in FACS buffer supplemented with propidium iodide. Mouse IgG1 (15H6, GeneTex) was used as an isotype control. The cell surface expression of DR5 was analyzed on a CytoFlex flow cytometer (Beckman Coulter, Brea, IN, USA).

### 4.10. Statistical Analysis

Cell culture experiments were repeated at least three times. The data were normally distributed and displayed as mean ± SD from at least three replicates. GraphPad Prism 8.0 (GraphPad Software Inc., San Diego, CA, USA) for Windows was used to generate graphical representations.

## 5. Conclusions

In conclusion, we have developed a high level of the heterologous production in *E. coli* of the bifunctional SRH–DR5-B protein with high cytotoxic activity both in 2D and 3D tumor cell culture models. The dual-targeted protein can exhibit strong anti-cancer activity with an improved therapeutic effect towards tumors of various origin with high levels of DR5 and VEGFR2 expression.

## Figures and Tables

**Figure 1 ijms-23-05860-f001:**
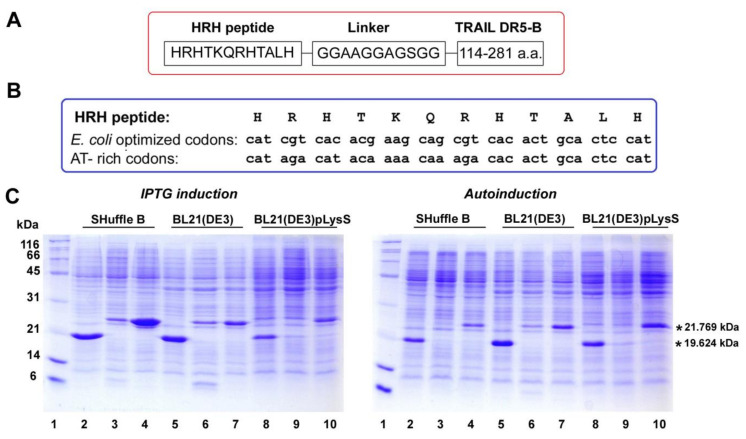
Scheme and optimization of HRH–DR5-B fusion expression in *E. coli* strains. (**A**) Scheme of the HRH–DR5-B fusion used in this study. (**B**) Codon modification of the HRH peptide to improve the expression of the fusion. (**C**) DNA enrichment with AT-rich codons encoding amino acids 2–7 of the HRH peptide significantly improved the expression of the HRH–DR5-B fusion. The expression of DR5-B and HRH–DR5-B was performed in *E. coli* strains SHuffle B, BL21(DE3) and BL21(DE3)pLysS by IPTG induction or autoinduction. Samples containing 10 μL of cell cultures were analyzed in 12% SDS PAGE. Lane 1—molecular weight markers; lanes 2, 5, 8—expression of DR5-B; lanes 3, 6, 9—expression of HRH–DR5-B with optimized codons for HRH peptide, and lanes 4, 7, 10—expression of HRH–DR5-B with AT-rich codons for HRH peptide. *—calculated mass of the monomeric DR5-B and HRH–DR5-B proteins 19.624 kDa and 21.769 kDa, respectively.

**Figure 2 ijms-23-05860-f002:**
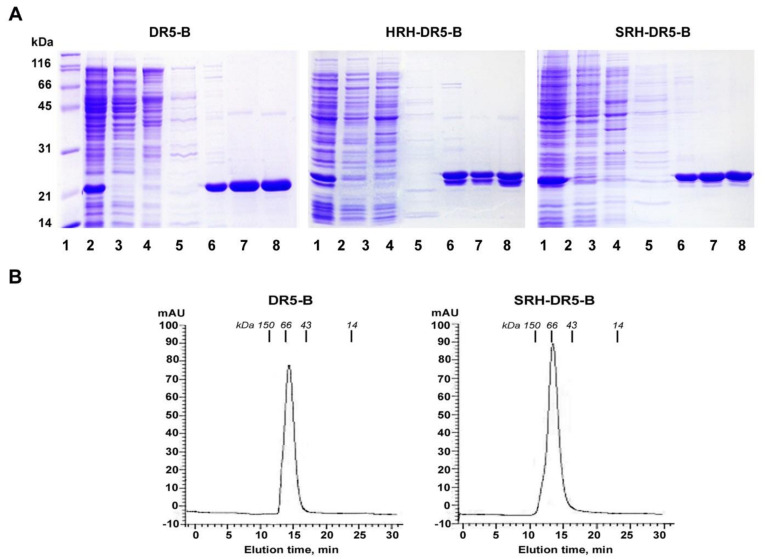
Purification of DR5-B, HRH–DR5-B and SRH–DR5-B proteins. (**A**) The proteins were purified from the soluble cytoplasmic fraction by metal affinity (on Ni-NTA agarose) and ion exchange (on SP Sepharose sorbent) chromatography. Samples containing 10–15 µg protein were analyzed on SDS-PAGE. Lane 1—molecular weight markers; lane 2—soluble fraction of cytoplasmic proteins; lanes 3–6 purification on Ni-NTA agarose; lane 3—fraction of proteins not associated with Ni-NTA agarose; lanes 4, 5—fraction of proteins washed with buffer containing 20 mM imidazole; lane 6—proteins eluted with a buffer containing 250 mM imidazole, lane 7—protein fractions after purification on the SP Sepharose sorbent; lane 8—samples of proteins after overnight dialysis. (**B**) Size-exclusion chromatography profile of DR5-B and SRH–DR5-B proteins performed on Superdex™ 75 10/300 GL column. IgG (150 kDa), BSA (66 kDa), ovalbumin (43 kDa) and lysozyme (14 kDa) were used as controls for the column calibration. The calculated mass of monomeric DR5-B and SRH–DR5-B was 19,624.04 Da and 21,769.42 Da, respectively. Accordingly, the molecular weight of trimeric DR5-B and SRH–DR5-B molecules is 58.872 kDa and 65.308 kDa, respectively.

**Figure 3 ijms-23-05860-f003:**
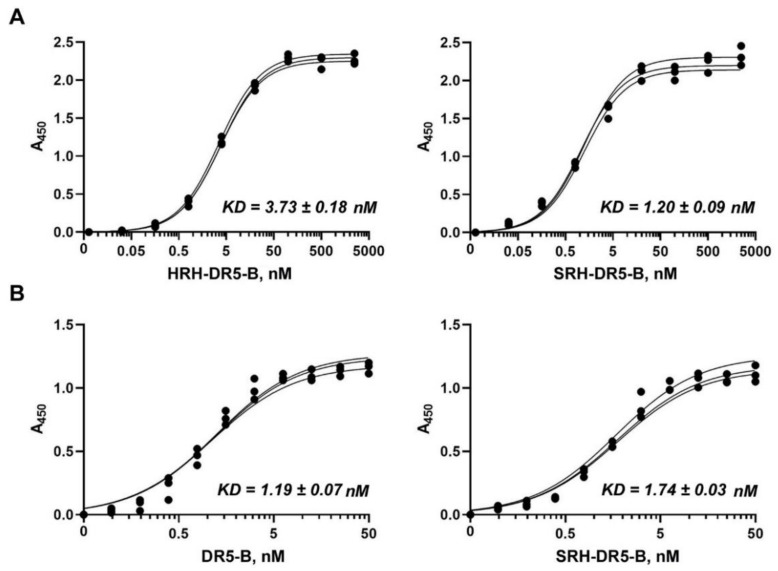
The binding affinity of SRH–DR5-B protein for receptors VEGFR2 and DR5 analyzed by enzyme-linked immunosorbent assay (ELISA). Ligands were added to the wells and pre-coated with VEGFR2 or DR5 receptors, and the binding was detected by anti-TRAIL monoclonal antibodies. (**A**) Comparison of the affinity of HRH–DR5-B and SRH–DR5-B fusions for the VEGFR2 receptor. (**B**) Comparison of the affinity of DR5-B and HRH–DR5-B proteins for the DR5 receptor. *KD* values were calculated from three independent experiments by the non-linear regression option in GraphPad Prism 8.0.

**Figure 4 ijms-23-05860-f004:**
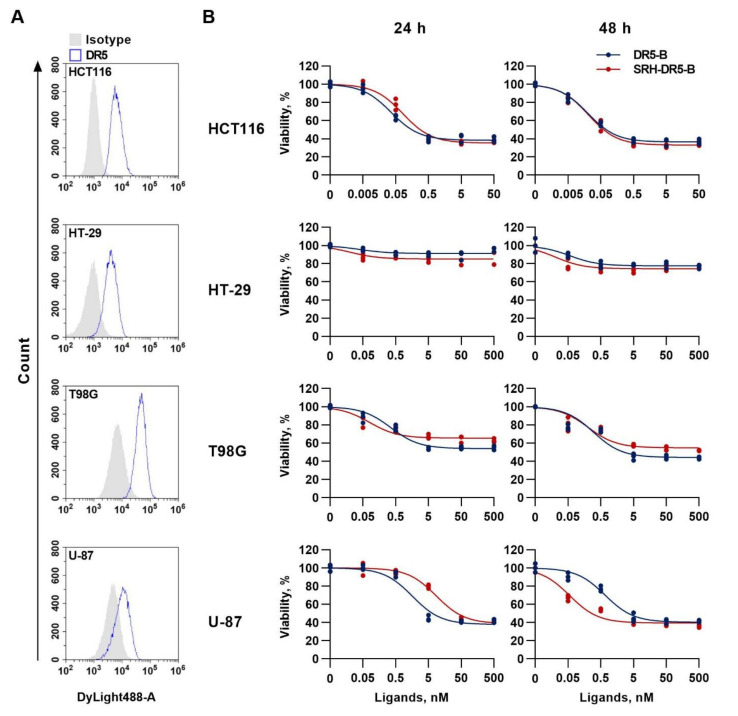
Cytotoxicity of DR5-B and SRH–DR5-B in 2D tumor cell models. (**A**) Surface expression of the DR5 receptor in tumor cells as determined by flow cytometry. (**B**) Human colorectal carcinoma (HT-29 and HCT116) and glioblastoma (T98G and U-87) cells were incubated with DR5-B or SRH–DR5-B for 24 and 48 h, and viability was analyzed by the WST-1 assay.

**Figure 5 ijms-23-05860-f005:**
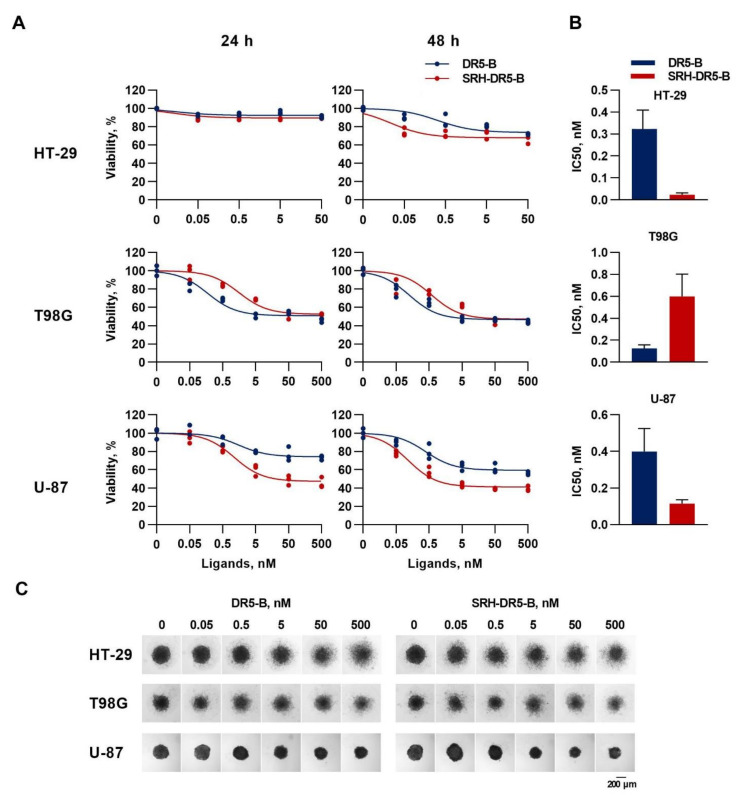
Cytotoxicity of SRH–DR5-B in a 3D culture of tumor cell spheroids. (**A**) HT-29, T98G and U-87 cell spheroids were generated in the agarose micro-wells after 48 h of cell seeding. Cell viability assay of 3D tumor spheroids after treatment with SRH–DR5-B and DR5-B for 24 h and 48 h, as determined by WST-1 test. **(B**) Ligand efficiency (IC50) calculated via nonlinear regression using GraphPad Prism 8.0. (**C**) Morphology of 3D tumor spheroids after 48 h incubation with ligands. The spheroids were imaged under 4× magnification using an inverted light microscope. Scale bar 200 μm.

## Data Availability

All data generated or analyzed during this study are included in this published article.

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
