# Peer review of "Optimized Heterologous Expression and Efficient Purification of a New TRAIL-Based Antitumor Fusion Protein SRH–DR5-B with Dual VEGFR2 and DR5 Receptor Specificity"

_ijms, 2022, doi:10.3390/ijms23115860_

Round 1

Reviewer 1 Report

The manuscript entitled " Optimized Heterologous Expression and Efficient Purification 2 of a New TRAIL-Based Antitumor Fusion Protein SRH-DR5-B 3
with Dual VEGFR2 and DR5 Receptor Specificity " was thoroughly reviewed. The work presented very well and clearly, and the results were impressive.  However, there are some questions to be clarified:

  1. In Materials and Methods, pls tell the total molecular weight of the fusion Protein SRH-DR5-B 3, and mark it in the SDS-PAGE. Pls explain  how the  11 amino acids fused with  TRAIL recognize the receptor.
  2.  In this work the His/Ser 270 substitution at the P2 position actually prevented protein degradation and did not affect the expression level.  please to explain the possible reasons in the Discussion section.

Author Response

Thank you very much for thoroughly reviewing our manuscript. We took into account all the comments and made the appropriate changes in the manuscript.

  1. In Materials and Methods, pls tell the total molecular weight of the fusion Protein SRH-DR5-B 3, and mark it in the SDS-PAGE. Pls explain how the  11 amino acids fused with  TRAIL recognize the receptor.

Answer: We have indicated the molecular weights of proteins in the Materials and Methods section, as well as in the caption to Fig. 1 and marked in Fig. 1.

“The calculated mass of monomeric DR5-B and SRH-DR5-B was 19624.04 Da and 21769.42 Da, respectively. Accordingly, the molecular weight of trimeric molecules DR5-B and SRH-DR5-B is 58.872 kDa and 65.308 kDa, respectively. We have included this information in the caption to Fig. 2 and in the text of results.

As an explanation for receptor recognition, we inserted the following explanations for SRH-DR5-B receptor binding in the results section "2.3 SRH-DR5-B showed high affinity for DR5 and VEGFR2 receptors".

“The synthetic anti-angiogenic peptide HRHTKQRHTALH was previously isolated from a phage display library using VEGFR-Fc as a bait [27]. The peptide did not lose its affinity for the VEGFR2 receptor after fusion with DR5-B at the N-terminus. This is probably due to the linker (GGAAGGAGSGG) between HRH and DR5-B, which provides free access of the peptide to the receptor.

To ensure that the addition of the N-terminal SRH peptide does not impair DR5 receptor binding, a comparative analysis of DR5-B and SRH-DR5-B proteins for DR5 binding was carried out by ELISA (Figure 3B). The affinity of SRH-DR5-B to the DR5 receptor was comparable to that of DR5-B (KD = 1.19 ± 0.07 nM and 1.74 ± 0.03 nM for DR5-B and SRH-DR5-B, respectively). It should be noted that the KD values of the DR5-B protein for the DR5 receptor, obtained by ELISA, almost coincided with our previous results obtained by the surface plasmon resonance (0.71 ± 0.013 nM) [28].

2. In this work the His/Ser 270 substitution at the P2 position actually prevented protein degradation and did not affect the expression level.  please to explain the possible reasons in the Discussion section.

Answer: We have inserted an explanation of the His/Ser substitution in the Discussion section.

“N-terminal methionine excision (NME) in bacteria is effective when the amino acids Gly, Ala, Pro, Ser, Thr, Val, and Cys are in position P2 [34]. It was also shown that a.a. Ala, Pro, Ser, Thr, Val, Cys and Lys at position P2 resulted in significantly higher expression of recombinant Igα than other amino acids [44]. Among them, the polar uncharged Ser (serine) was chosen to replace His at the P2 position. As a result, the His/Ser substitution did not affect the expression level and virtually prevented protein degradation.”

New citation:

  1. Bivona, L.; Zou, Z.; Stutzman, N.; Sun P.D. Influence of the second amino acid on recombinant protein expression Protein Expr Purif 2010, 74, 248–256, doi: 10.1016/j.pep.2010.06.005

Reviewer 2 Report

Research Article: “Optimized Heterologous Expression and Efficient Purification of a New TRAIL-Based Antitumor Fusion Protein SRH-DR5-B with Dual VEGFR2 and DR5 Receptor Specificity.”

In this manuscript authors have designed the HRH-DR5-B fusion protein based on the DR5-selective mutant variant of TRAIL fused to the anti-angiogenic synthetic peptide HRHTKQRHTALH. HRH-DR5-B fusion protein expression in E. coli was enhanced by the substitution of E. coli-optimized codons with AT-rich codons in the DNA sequence encoding the first 7 amino acid residues of the HRH peptide. Furthermore, replacement of His by Ser at position P2 immediately after the initiator Met dramatically minimized degradation that allowed more than 20 mg of protein to be obtained from 200 ml of cell culture. Authors have concluded that SRH-DR5-B fusion bound the VEGFR2 and DR5 receptors with high affinity and showed increased cytotoxic activity in 3D multicellular tumor spheroids. This article is well planned, well executed and well written but this reviewer has few comments that can improve the quality of the manuscript.  

1, The English of manuscript can be polished (minor) and there are few typo errors in the manuscript that can be checked.

2, At least one illustrative figure may be provided as to highlight the summary of this study.

3, Authors can add one paragraph for abbreviations.

4, Did authors have checked any immunogenic potential of SRH-DR5-B for in vivo experiments?

5, Authors can discuss the limitations to their study.

Author Response

Thank you very much for thoroughly reviewing our manuscript. We took into account all the comments and made the appropriate changes in the manuscript.

1, The English of manuscript can be polished (minor) and there are few typo errors in the manuscript that can be checked.

Answer: We checked English in detail and corrected typos.

2, At least one illustrative figure may be provided as to highlight the summary of this study.

Answer: We made a graphic abstract to highlight the summary of this study.

3, Authors can add one paragraph for abbreviations.

Answer: There is no requirement for abbreviations in the rules for the author of this journal. Anyway, we created Table S1 for abbreviation and putted it in supplementary materials.

4, Did authors have checked any immunogenic potential of SRH-DR5-B for in vivo experiments?

Answer: We plan to investigate the antitumor properties of SRH-DR5-B in human cancer xenograft models, in particular colorectal cancer and glioblastoma tumors.

5, Authors can discuss the limitations to their study.

Answer: A very reasonable suggestion. We have added a paragraph about potential limitations in the discussion section.

“Of course, any bifunctional protein preparation may have potential limitations or problems in clinical use. BFPs that act on different receptors on tumor cells or block two signaling pathways at the same time are intended primarily for the treatment of patients with solid tumors, but most of them have not yet been approved for clinical use. Limitations may be related to various factors, such as drug yield and cost, specificity, half-life, ability to penetrate the tumor and adverse effects. The technology developed by us provides a high yield of the SRH-DR5-B, which demonstrated a high specificity for the DR5 and VEGFR2 receptors. Further preclinical studies will show the antitumor efficacy as well as the potential limitations of the drug in order to move it into clinical trials.”